# Voriconazole Use in Children: Therapeutic Drug Monitoring and Control of Inflammation as Key Points for Optimal Treatment

**DOI:** 10.3390/jof7060456

**Published:** 2021-06-07

**Authors:** José María Valle-T-Figueras, Berta Renedo Miró, Maria Isabel Benítez Carabante, Cristina Díaz-de-Heredia, Jaume Vima Bofarull, Natalia Mendoza-Palomar, Maria Teresa Martín-Gómez, Pere Soler-Palacín

**Affiliations:** 1Paediatric Infectious Diseases and Immunodeficiencies Unit, Hospital Universitari Vall d’Hebron, Universitat Autònoma de Barcelona, 08035 Barcelona, Catalonia, Spain; jvalle@santpau.cat (J.M.V.-T.-F.); nataliamendoza@upiip.com (N.M.-P.); 2Department of Paediatrics, Hospital Universitari de la Santa Creu i Sant Pau, Universitat Autònoma de Barcelona, 08041 Barcelona, Catalonia, Spain; 3Pharmacy Department, Hospital Universitari Vall d’Hebron, 08035 Barcelona, Catalonia, Spain; brenedo@vhebron.net; 4Department of Paediatric Haematology and Oncology, Hospital Universitari Vall d’Hebron, Universitat Autònoma de Barcelona, 08035 Barcelona, Catalonia, Spain; mbenitez@vhebron.net (M.I.B.C.); crdiaz@vhebron.net (C.D.-d.-H.); 5Department of Clinical Biochemistry. Central Clinical Laboratories, Hospital Universitari Vall d’Hebron, 08035 Barcelona, Catalonia, Spain; jvima@vhebron.net; 6Microbiology Department, Hospital Universitari Vall d’Hebron, Universitat Autònoma de Barcelona, 08035 Barcelona, Catalonia, Spain; mtmartin@vhebron.net

**Keywords:** paediatric fungal infections, antifungal treatment, therapeutic drug monitoring, voriconazole, children, inflammation

## Abstract

Voriconazole plasma concentrations (PC) are highly variable, particularly in children. Dose recommendations in 2–12-year-old patients changed in 2012. Little data on therapeutic drug monitoring (TDM) after these new recommendations are available. We aimed to evaluate voriconazole monitoring in children with invasive fungal infection (IFI) after implementation of new dosages and its relationship with safety and effectiveness. A prospective, observational study, including children aged 2–12 years, was conducted. TDM was performed weekly and doses were changed according to an in-house protocol. Effectiveness, adverse events, and factors influencing PC were analysed. A total of 229 PC from 28 IFI episodes were obtained. New dosing led to a higher rate of adequate PC compared to previous studies; still, 35.8% were outside the therapeutic range. In patients aged < 8 years, doses to achieve therapeutic levels were higher than recommended. Severe hypoalbuminemia and markedly elevated C-reactive protein were related to inadequate PC. Therapeutic PC were associated with drug effectiveness and safety. Higher doses in younger patients and a dose adjustment protocol based on TDM should be considered. Voriconazole PC variability has decreased with current updated recommendations, but it remains high and is influenced by inflammatory status. Additional efforts to control inflammation in children with IFI should be encouraged.

## 1. Introduction

Invasive fungal infection (IFI) is an uncommon condition, occurring almost exclusively in immunosuppressed and critically ill patients. It is associated with high morbidity and mortality, especially when caused by filamentous fungi [1,2,3,4,5,6]. Management of IFI in children requires considerable expertise, and the appropriate diagnosis and treatment can determine the course of the disease. Approved in 2002 by the FDA, voriconazole is a broad-spectrum triazole antifungal agent that has become a mainstay of both primary and salvage treatment for invasive aspergillosis and candidiasis [1,7,8,9,10,11,12,13,14]. Several publications, mainly in adults but also in children, suggest that therapeutic drug monitoring (TDM) improves the efficacy and safety of voriconazole [1,8,15,16,17,18,19,20,21,22,23,24]. As between-subject differences in voriconazole plasma concentrations (PC) are more marked in children than in adults [25,26,27,28,29,30], monitoring of trough concentrations is particularly important in children to ensure optimal treatment.

In 2011, our group reported a relationship between sub-therapeutic voriconazole concentrations and lower treatment effectiveness in paediatric patients, and between supra-therapeutic concentrations and neurological and skin toxicity. In addition, we suggested the need to increase dosage in younger patients to achieve appropriate PC [17]. In 2012, based on a pharmacokinetic study by Friberg et al. [31], the recommended voriconazole dose was changed in patients aged 2 to 12 years, with an increase in the maintenance dose to 16 mg/kg/day. Since that time, only a small number of studies have evaluated voriconazole monitoring in children according to the new recommendations, and most are in Asian populations [32,33,34]. The aim of this study was to assess the variability of plasma voriconazole in children receiving the drug according to the current posology for treating IFI. The secondary aim was to determine the overall effectiveness and safety of voriconazole treatment and investigate relationships between these aspects and the drug monitoring findings.

The findings of this study indicate that variability of voriconazole remains high despite the 2012 updated dosing recommendations, and that drug levels are influenced by severe hypoalbuminemia and elevated C-reactive protein (CRP). Voriconazole at appropriate doses is effective and safe in children, and PC are related to treatment effectiveness and safety in children with IFI. Nonetheless, higher voriconazole doses should be considered in patients below 8 years old, and standardized recommendations for dose adjustment should be designed based on monitoring results.

## 2. Materials and Methods

This prospective study was conducted at Vall d’Hebron Children’s Hospital, a tertiary-care referral centre in Barcelona (Catalonia, Spain) with an average of 7500 hospitalizations per year and a dedicated paediatric cancer and hematopoietic stem cell transplantation (HSCT) department. Around 250 paediatric cancer patients are treated and 40 HSCT procedures are performed per year in our centre.

### 2.1. Ethics

The study was approved by the local ethics committee (EPA (AMI) 69/2013 (3874)). All procedures were in accordance with the 1964 Helsinki declaration and its later amendments. Informed consent was obtained from the legal representatives of all participating patients.

### 2.2. Inclusion Criteria

The study included consecutive patients aged 2 to 12 years and weighing less than 50 kg, who had received medical care at our centre from January 2014 to August 2018 due to proven, probable, or possible IFI, established according to the diagnostic criteria of the European Organization for Research and Treatment of Cancer/Invasive Fungal Infections Cooperative Group and the National Institute of Allergy and Infectious Diseases Mycoses Study Group (EORTC/MSG) Consensus Group [35]. In 2019, the EORTC/MSG definitions were updated [36], but the update was not used in the present study, as data collection ended in 2018. Patients were required to have received voriconazole as treatment for IFI with the 2012 recommended dosage according to the datasheet for this age group.

### 2.3. Procedures

Intravenous voriconazole administration consisted of an initial loading dose of 9 mg/kg q12h in the first 24 h followed by a maintenance dose of 8 mg/kg q12h. Oral administration used a dose of 9 mg/kg q12h without a loading dose.

Voriconazole trough levels were determined within the first week of treatment and weekly thereafter using a Shimadzu Nexera HPLC system with a fluorescence detector for antifungals [37,38]. The lower limit of quantification, defined as the minimum voriconazole concentration measured in plasma with an accuracy of around 5%, was 0.3 mg/L. Values below this limit were recorded as <0.3 mg/L. Voriconazole concentrations of 1 to 5.5 mg/L (both values included) were considered adequate regardless of the location of the infection.

When voriconazole levels were outside the desired therapeutic range, a dose adjustment recommendation was determined following the local protocol used in the study: at values <1 mg/L, a 50% daily dose increase was recommended; at values >5.5 mg/L, voriconazole was discontinued for 24 h, followed by a 50% daily dose decrease. When adverse events (AEs) attributable to voriconazole were considered relevant by the attending clinician, voriconazole was stopped until PC dropped to within the desired range and AEs were considered controlled; voriconazole was then reinitiated with a 50% decrease in daily dose.

### 2.4. Data Collection and Definitions

Epidemiological and clinical data, complementary examinations, relevant co-medications in terms of interactions, and concomitant antifungal medication were recorded. Follow-up continued throughout the treatment period (including maintenance treatment) up to at least 6 months after the start of voriconazole in surviving patients. At least 2 treatment response evaluations were performed (early and late evaluation) in accordance with the EORTC/MSG treatment response criteria [39]. Complete and partial response were defined as successful treatment and any other situation was defined as treatment failure, including deaths that were not attributable to IFI.

AEs were classified according to the Common Terminology Criteria for Adverse Events v5.0 (U.S. Department of Health and Human Services, National Institutes of Health, National Cancer Institute) [40]. Abnormalities detected during treatment that were not present before starting treatment, and those that were present and worsened during treatment were considered as possibly related to voriconazole.

### 2.5. Statistical Analysis

Several statistical tests were carried out to analyse relationships between plasma voriconazole concentrations and the study variables. Descriptive statistics were used to summarize data related to the patients, IFI episodes and adverse events: the median and interquartile range (IQR) for quantitative variables and the frequency for qualitative ones. The chi-square test or likelihood ratio test, as appropriate, were performed for qualitative variables. The Student’s *t*-test (or ANOVA) or the Wilcoxon test (or Kruskal–Wallis test), as appropriate, were conducted for quantitative variables. The Pearson correlation coefficient was used to study relationships between two quantitative variables.

Only proven or probable IFI episodes were included in the analysis of treatment effectiveness, whereas data from all IFI episodes were included in the safety analysis. Considering the dichotomous nature of the secondary dependent variables—treatment effectiveness (success or failure), AEs (presence or absence), and plasma voriconazole value (≤5.5 mg/L or >5.5 mg/L)—three multivariate logistic models (MLM) were developed, including all covariates in the models. Voriconazole values were categorized into three levels: sub-therapeutic, therapeutic, and supra-therapeutic. In this case, a multinomial model was used. In all models, a backward procedure was applied to eliminate covariates from the analysis based on statistical criteria.

Finally, a cut-off point was established to form two groups (<8 y and ≥8 y), using the age at which significant differences were seen in the dose administered to achieve correct PC. To analyse the effect of dose on the voriconazole level, a repeated measures analysis using a linear mixed-effects model (LMM) was applied in each group: patients were considered random factors and the various dose determinations were the within-subject repeated measures. For this analysis, we excluded patients receiving medication that could significantly change voriconazole concentrations according to the datasheet: carbamazepine, phenobarbital, efavirenz, rifabutin, rifampicin, ritonavir, St. John’s wort, fluconazole, and phenytoin.

The statistical analysis was performed using the SAS System v9.4 (SAS Institute Inc., Cary, NC, USA). A nominal significance level of 5% (*p* < 0.05) was applied in all statistical tests.

## 3. Results

### 3.1. Characteristics of Patients and Invasive Fungal Infections

Twenty-seven patients were included, 55% male and a median age of 9 years (IQR 6–10 y). Most patients (85.2%) were non-Asian and mainly white Europeans (59%) (Table 1). The number of patients in each ethnic group was not large enough to evaluate potential differences according to ethnicity. Overall, 28 IFI episodes were recorded (18 proven/probable) with the predominant cause being filamentous fungi, particularly *Aspergillus* spp. The lung was the most commonly affected site, followed by the cranial sinus and central nervous system; a total of 28.6% of patients had disseminated disease (Table 2). The median duration of voriconazole treatment in patients with proven or probable IFI was 80.5 days (IQR 15–117 days). Although a dual antifungal regimen (voriconazole plus liposomal amphotericin B) was used at some point to treat 12 of the 18 episodes of proven/probable IFI, the median duration of this treatment was 27.5 days (12.8–64.3), approximately one-third the total duration of voriconazole treatment (Table 2).

### 3.2. Therapeutic Drug Monitoring

Two hundred and twenty-nine voriconazole determinations were analysed (Table 3), and 64.2% were within the therapeutic range. Among those outside the range (35.8%), 27.5% were sub-therapeutic and 8.3% supra-therapeutic. The percentage of sub-therapeutic levels during the first week of treatment (43.5%) was higher than in the remainder of the treatment period (25.9%), but the difference was not statistically significant (chi-square, 3.21; *p* = 0.07).

Dose changes corrected 63% of determinations outside the therapeutic range. This percentage rose to 75% when the protocolled dose adjustment recommendations were strictly adhered to, although the difference was not statistically significant (chi-square, 2.38; *p* = 0.12).

To achieve therapeutic concentrations, the estimated average dose was significantly higher in patients <8 years than in those ≥8 years (21 vs. 16.5 mg/kg/day; LMM, F = 5.16; *p* = 0.02). A significant effect of the dose administered on the voriconazole concentration was observed in the younger group (LMM, F = 4.79; *p* = 0.01): a dose increase of one unit increased the plasma concentration by 0.08 units.

Regarding factors that could potentially affect voriconazole levels, no differences were found in association with the administration route or concomitant omeprazole administration. However, statistically significant differences were observed when albumin was <3 mg/dL (chi-square, 12.04; *p* = 0.02) (Table 3).

In addition, this analysis was performed after grouping the voriconazole levels into two categories (therapeutic and non-therapeutic). In this case, CRP values ≥ 4 mg/dL (chi-square, 4.85; *p* = 0.03) and albumin < 3 mg/dL (chi-square, 7.31; *p* = 0.03) were both significantly associated with inadequate PC (Table 4).

### 3.3. Effectiveness

Treatment effectiveness in the early evaluation was 60%. This value increased to 73.3% when stabilized filamentous IFI in a severely immunosuppressed patient was considered a treatment success, as is suggested in the EORTC/MSG treatment response criteria [39]. Treatment effectiveness in the late evaluation was 53%. When the late assessment included a 6-month evaluation in patients receiving salvage treatment for IFI due to filamentous fungi, the overall late effectiveness rate rose to 60%. No breakthrough infections occurred during the study period. Six-month survival was 80%, and no deaths were attributed to IFI (Table 5).

In the early evaluation, the presence of graft-versus-host disease (chi-square, 6.96; *p* < 0.01) and use of dual antifungal therapy (chi-square, 5.04; *p* = 0.02) were both associated with a poorer disease course. The late evaluation revealed a significant relationship between voriconazole levels ≥1 mg/L and treatment effectiveness (chi-square, 7.16; *p* = 0.03).

### 3.4. Safety

Liver enzyme abnormalities were the most common AEs (24/27 patients), with increases in gamma glutamyl transferase (GGT) values predominating over alanine aminotransferase (ALT), total bilirubin (TBIL), or alkaline phosphatase (ALP) changes. However, most AEs were not severe (grade < 3), had no clinical impact, and reversed at completion of treatment. Non-hepatic AEs were much less common, and all were grade one or two. These included gastrointestinal, neurological, or cutaneous AEs (2 of each in the 27 patients) and visual or renal AEs (1 of each in the 27 patients) (Table 6). Voriconazole withdrawal was required in two AEs (7.1%): one case of photosensitivity and another of gastrointestinal intolerance to oral voriconazole.

A statistically significant relationship was observed between supra-therapeutic voriconazole concentrations and abnormally increased ALP levels (chi-square, 5.91; *p* = 0.02). Although TBIL and GGT elevations were also more frequent at high voriconazole levels, the associations were not significant (chi-square, 0.27; *p* = 0.6 and chi-square, 1.66; *p* = 0.2, respectively). Finally, there was a significant relationship between supra-therapeutic voriconazole and renal toxicity (chi-square, 4.81; *p* = 0.03).

Multivariate analysis showed no significant associations of any of the variables studied with plasma voriconazole variability or treatment effectiveness, except a marginally significant relationship between normal or only slightly elevated CRP and adequate plasma drug concentrations (MLM, F = 3.22, *p* = 0.07). There were, however, associations between hypoalbuminemia and increased TBIL (MLM, F = 4.96, *p* = 0.03), oral administration and increased ALT (MLM, F = 4.09, *p* = 0.04), and a statistical trend to ALT elevation when dual antifungal therapy was used (MLM, F = 3.75, *p* = 0.054).

## 4. Discussion

The findings of this study indicate that voriconazole PC, within the therapeutic range, are related to treatment effectiveness and safety in children with IFI. In addition, they show that variability of the drug remains high despite the 2012 updated dosing recommendations, and that voriconazole levels are influenced by severe hypoalbuminemia and elevated CRP.

Information on voriconazole TDM in children is scarce, particularly in non-Asian populations and using the new dosage recommendations [31,32,33,34,41,42,43], and the available data are often heterogeneous and difficult to compare. The main paediatric studies focussed on plasma voriconazole variability and monitoring published since 2012 are summarized in Table 7. Most of these studies include a relatively small sample (11 to 74 patients) [17,29,30,32,33,34,41,44,45,46,47,48,49]. An exception is the article by Liu et al. [50] with 107 patients (75 younger than 2 y) and the study by Allegra et al. [51] with 237 patients, although 150 received voriconazole as prophylaxis. As in most of these studies, our sample is relatively small, but enrolment was prospective and under strict criteria, such as a narrow age range and exclusion of patients receiving voriconazole as primary prophylaxis. Our patient population predominantly included white Europeans (59.3%), with a minority of Asians (14.8%). It is reported that genetic polymorphisms conditioning voriconazole metabolism may differ depending on the patients’ ethnicity, and this is especially relevant in Asian patients [27,52,53,54,55]. Many of the paediatric studies cited above included a predominantly Asian population or did not describe the patients’ ethnicity [30,32,41,44,45,46,47,48,49,50].

The type of patients and IFIs included here (mainly haematological malignancies and HSCT, and a predominance of *Aspergillus* spp.) are consistent with reported data [17,29,30,32,41,45,46,47,48,49], as are the affected sites and percentage of disseminated infections (28.6%) [10]. The median duration of voriconazole treatment in our study, almost 12 weeks (80.5 days), is longer than values reported in most of the recent paediatric studies, which range from 23.5 to 72 days [17,29,32,41,45,46,49], although there are some exceptions, such as the study by Tucker et al. [48] (105 days, 11 patients) and Lempers et al. [33] (118 days, 21 patients). As was mentioned, our follow-up included voriconazole maintenance treatment; no specific data on this are available in most of the studies cited.

As is seen in Table 7, voriconazole PC often fell outside the therapeutic range. Adequate levels were described in 34% to 50% of patients in older studies using the previous dosage [17,29,30,44,45,46,47,50]. With the new dosing recommendations, our findings indicate an improvement in this regard. The percentage of patients with therapeutic concentrations was significantly higher in the present study than our previous one [17] (chi-square, 19.38; *p* < 0.01), but voriconazole variability remained high. Of note, some studies have shown that voriconazole PC are less adequate at the beginning of treatment [33,45], a trend we also observed. In addition, some authors report the need for higher doses than recommended to achieve optimal levels, especially in younger patients [17,30,33,41,46,49,56], which agrees with our finding that patients younger than 8 years required a considerably higher dose than is recommended in the datasheet. According to our data, a maintenance dose increase that reaches 10 to 11 mg/kg q12h in patients aged 2 to 7 years would lead to more appropriate voriconazole levels.

Almost two-thirds of inadequate plasma voriconazole values in the present study were corrected with dose changes. However, sometimes the protocol recommendations were not strictly followed. In some instances, we found no clinical justification for non-compliance, but in most cases the PC values were very close to the therapeutic interval and the attending physician decided to make a smaller dose change than was recommended. Although a statistically significant difference was not obtained, the percentage of PC correction was higher when the study recommendations were strictly followed (Figure 1). Other studies have reported a correction rate of around 60% to 80%, but none of them have described a predefined scheme for dose adjustment [30,33,41]. Predefined recommendations for adjusting voriconazole dose following monitoring may favour early correction of voriconazole levels and should be standardized.

Of note, we found that CRP and albumin status had an impact on plasma voriconazole, with less adequate concentrations in patients with significantly elevated CRP and markedly decreased albumin. In adults, systemic inflammation has been suggested to play a role in voriconazole variability [57,58,59,60,61,62,63,64]. To our knowledge, only the studies by ter Avest et al. [65], Luo et al. [66] and Liu et al. [50] have described a similar impact in paediatric patients. Our findings indicate that controlling systemic inflammation and optimizing albumin values could significantly contribute to enhancing the quality of voriconazole treatment in this population.

As to the effectiveness of voriconazole with the new dosage, our results are similar or slightly better than those reported in previous paediatric studies [17,29,45,46,48,49]. Nevertheless, it is difficult to perform a direct comparison, as most of these reports did not use the EORTC/MSG treatment response criteria [39], some included use of the drug in prophylaxis or in possible IFI, and others excluded deaths not attributable to IFI [17,29,32,34,41,45,46,48,50].

Dual antifungal therapy was often used in the present study (mainly to cover a period of possibly inadequate voriconazole PC), but it was of limited duration and not associated with clinical benefits. Based on our experience, we believe that dual antifungal therapy use should be limited to the first days of treatment, until voriconazole PC reach the recommended range.

The relationship between adequate plasma voriconazole levels and treatment effectiveness can be considered validated in adults [18,22,23,28,67,68,69,70], but the evidence in paediatrics is relatively poor [17,45,49,71]. We found that concentrations higher than 1 mg/L were the only factor related to an improvement in treatment effectiveness at the later evaluation. To our knowledge, the two studies by our group are the only ones demonstrating this relationship in non-Asian paediatric patients and using the EORTC/MSG treatment response criteria.

The available paediatric data on voriconazole safety are heterogeneous, as the criteria for classifying AEs and attributing them to the drug vary widely between studies. However, the percentage of AEs we detected is similar to the values reported in other studies using the same AE classification [29,44,45,46,72]. We mention that our recording of up to four liver function parameters, including changes with no clinical impact, may have led to an overestimation of hepatic AEs in our study. Voriconazole-attributed toxicity led to discontinuation of the drug in a small number of cases (7.1%), comparable to the lower range in previous paediatric studies, which report percentages of 3.2% to 26% [29,30,32,41,43,45,48,73]. The relationship between plasma voriconazole levels and treatment safety has been demonstrated in adults, although there are some inconsistencies regarding the type of AEs and the appropriate cut-off point to indicate a risk of toxicity [18,22,23,28,68,69,70]. Again, in paediatrics there is much less evidence in this regard [17,32]. Supra-therapeutic concentrations were associated with neurological and skin AEs in our previous study [17], and with liver AEs in the study by Martin et al. [32]. We found a relationship between elevated plasma voriconazole and liver and kidney AEs. Although the statistical reliability in our analysis of renal AEs was limited by the small number of cases detected, this association has also been reported by Chu et al. [74] in adults.

The present study has several limitations. First, the relatively small sample size constrained the statistical power of some analyses, as also occurred in many of the previous studies. Second, the patients’ high comorbidity and administration of dual antifungal drugs in some episodes were potential confounding factors when interpreting treatment safety and effectiveness, also a common point in previous studies. Third, the role of genetic polymorphisms of cytochrome P450 was not analysed, as no clear recommendations can be determined on the basis of this finding. Despite these limitations, we believe the study provides new data regarding voriconazole use in the treatment of IFI in the paediatric population that will facilitate comparison with those reported in other studies.

## 5. Conclusions

Voriconazole at appropriate doses is effective and safe in children. Nonetheless, our results show that plasma voriconazole variability in paediatric IFI patients remains high despite the updated dosage recommendations. Higher voriconazole doses are needed in patients younger than 8 years, and standardized recommendations for dose adjustment should be designed based on monitoring results. Considering the relationship found between appropriate PC and voriconazole effectiveness and safety in children, TDM should be mandatory in this population. In addition to monitoring, efforts should be intensified to control inflammation, as this factor has an impact on achieving therapeutic drug levels.

## Figures and Tables

**Figure 1 jof-07-00456-f001:**
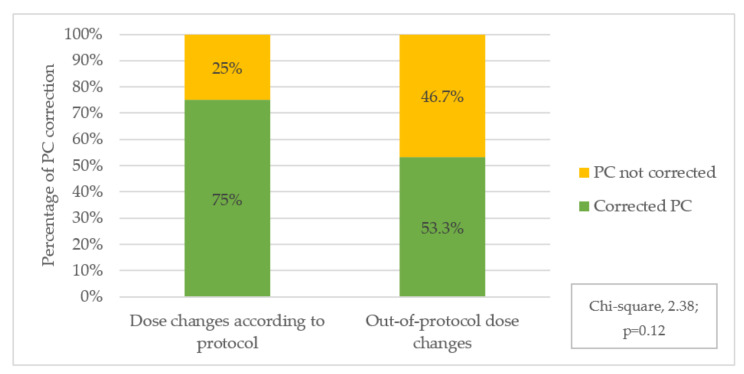
Voriconazole plasma concentrations (PC) corrections after dose changes according to protocol compliance.

**Table 1 jof-07-00456-t001:** Baseline patient characteristics.

		N (%) or Median (IQR)
Patients		27
Age, y		9 (6–10)
Weight, kg		25 (20.2–32.3)
Sex	Male	15 (55.6)
	Female	12 (44.4)
Ethnicity	White	16 (59.3)
	Asian	4 (14.8)
	Maghreb	3 (11.1)
	South American	3 (11.1)
	Sub-Saharan	1 (3.7)
Underlying diseases	PID ^a^	8 (29.6)
	ALL	8 (29.6)
	AML	5 (18.5)
	MDS	1 (3.7)
	Other ^b^	5 (18.5)
Previous allogenic HSCT	Yes	18 (66.7)
	No	9 (33.3)
Previous GVHD (in HSCT patients)	Yes	10 (55.6)
	No	8 (44.4)
Neutropenia at the beginning of treatment	Yes	12 (44.4)
	No	15 (55.6)
CMV viremia during IFI episode (in HSCT patients)	Yes	8 (44.4)
	No	10 (55.6)

Abbreviations: ALL, acute lymphocytic leukaemia; AML, acute myeloid leukaemia; CMV, cytomegalovirus; GVHD, graft-versus-host disease; HSCT, haematopoietic stem cell transplantation; IFI, invasive fungal infection; MDS, myelodysplastic syndrome; PID, primary immunodeficiencies. ^a^—4 chronic granulomatous disease, 3 severe combined immunodeficiency, 1 hyper-IgM syndrome. ^b^—3 medullary aplasia, 1 major thalassemia, 1 Fanconi anaemia.

**Table 2 jof-07-00456-t002:** Characteristics of proven or probable IFI episodes.

			N (%) or Median (IQR)
Episodes of proven or probable IFI ^a^			18
Microorganism	Moulds	*Aspergillus* spp. ^b^	13 (72.2)
		*Cladosporium cladosporioides*	1 (5.6)
		*Fusarium solani*	1 (5.6)
	Yeasts	*Candida* spp. ^c^	2 (11.1)
		*Trichosporon asahii*	1 (5.6)
Affected sites ^d^	Disseminated ^d^		6 (33)
	Lung		13 (72.2)
	Cranial sinus		2 (11.1)
	CNS		2 (11.1)
	Skin		3 (16.7)
	Bloodstream		1 (5.6)
	Spleen		2 (11.1)
	Kidney		2 (11.1)
	Heart (endocarditis)		2 (11.1)
	Other sites (8 cases) ^e^		1 (5.6)
Type of treatment	Primary		10 (55.6)
	Salvage treatment		5 (27.8)
	Suppressive treatment ^f^		3 (16.7)
Duration of voriconazole treatment, days			80.5 (15–117)
Dual antifungal therapy at some point in the episode			12 (66.7)
Duration of dual antifungal treatment, days			27.5 (12.8–64.3)
Surgery			10 (55.6)

Abbreviations: CNS, central nervous system; IFI, invasive fungal infection. ^a^—Based on the diagnostic criteria of the European Organization for Research and Treatment of Cancer/Invasive Fungal Infections Cooperative Group and the National Institute of Allergy and Infectious Diseases Mycoses Study Group Consensus Group. ^b^—3 *A. fumigatus*, 2 *A. flavus*, 1 *A. nidulans*, 4 *Aspergillus* sp. (unidentified *Aspergillus* isolates), 5 probable *Aspergillus* spp. (clinical suspicion and positive serum galactomannan test). ^c^—1 *C. albicans*, 1 *C. krusei*, 1 *C. lusitaniae.*
^d^—6 episodes involved more than 1 site. ^e^—Bone, skin and mucosa with angioinvasion, eyes, liver, retropharyngeal abscess, venous thrombosis and neck soft tissue. ^f^—Antifungal therapy once the acute phase is over and clinical stability is verified.

**Table 3 jof-07-00456-t003:** Analysis of plasma voriconazole levels during treatment.

		Plasma Voriconazole Levels			*p*
		<1 mg/L	1–5.5 mg/L	>5.5 mg/L	
Determinations, *n* (%)		63 (27.5)	147 (64.2)	19 (8.3)	-
Dose, mean (min-max) ^a^		18.4 mg/kg/day (3.7–52)	18.2 mg/kg/day (5.4–40)	18.1 mg/kg/day (4.5–38)	NS
Administration route	IV, *n* (%)	41 (29.1)	87 (61.7)	13 (9.2)	NS
	Oral, *n* (%)	22 (25)	60 (68.2)	6 (6.8)	
Concomitant omeprazole administration	No, *n* (%)	7 (22.6)	20 (64.5)	4 (12.9)	NS
	Yes, *n* (%)	56 (28.3)	127 (64.1)	15 (7.6)	
CRP ^b^	≤4 mg/dL, *n* (%)	31 (23.3)	92 (69.2)	10 (7.5)	NS
	>4 mg/dL, *n* (%)	26 (34.2)	41 (54)	9 (11.8)	
Plasma albumin	>3.4 mg/dL, *n* (%)	28 (23.5)	83 (69.8)	8 (6.7)	0.02
	3–3.4 mg/dL, *n* (%)	21 (30.4)	45 (65.2)	3 (4.4)	
	<3 mg/dL, *n* (%)	14 (34.2)	19 (46.3)	8 (19.5)	

Statistical inference using the chi-square test. Abbreviations: CRP, C-reactive protein; IV, intravenous; NS, not significant. ^a^—Average dose administered taking into account changes in the initial dosage depending on plasma values of the drug. ^b^—Available in 209/229 determinations.

**Table 4 jof-07-00456-t004:** Relationship between adequate voriconazole level and CRP and albumin status.

		Plasma Voriconazole Levels			*p*
		Overall	Non-Therapeutic (<1 mg/L; >5.5 mg/L)	Therapeutic (1–5.5 mg/L)	
Determinations, *n* (%)		229	82 (35.8)	147 (64.2)	-
CRP ^a^	≤4 mg/dL	133 (63.6)	41 (30.8)	92 (69.2)	0.03
	>4 mg/dL	76 (36.4)	35 (46)	41 (54)	
Plasma albumin	>3.4 mg/dL	119 (52)	36 (30.3)	83 (69.7)	0.03
	3–3.4 mg/dL	69 (30.1)	24 (34.8)	45 (65.2)	
	<3 mg/dL	41 (17.9)	22 (53.7)	19 (46.3)	

Statistical inference using the chi-square test. Abbreviations: CRP, C-reactive protein. ^a^—Available in 209/229 determinations.

**Table 5 jof-07-00456-t005:** Response to treatment in proven or probable IFI episodes.

Response to Treatment ^a^, *n* = 15 ^b^					
Early evaluation (4–6 weeks) ^c^	Success, *n* (%)	9 (60)	Complete response		3 (20)
			Partial response		6 (40)
	Failure, *n* (%)	6 (40)	Stable disease		4 (26.7)
			Progression of IFI		2 (13.3)
			Death		-
Late evaluation (12 weeks)	Success, *n* (%)	8 (53.3)	Complete response		5 (33.3)
			Partial response		3 (20)
	Failure, *n* (%)	7 (46.7)	Stable disease		1 (6.7)
			Progression of IFI		4 (26.7)
			Death		2 (13.3)
Survival at 6 months, *n* (%)				12 (80)	
Death attributed to IFI/death due to any cause at 6 months, *n*/*n*				0/3 ^d^	
Follow-up, median (IQR)				326 days (177–614.5)	

Abbreviations: IFI, invasive fungal infection. ^a^—Response to therapy according to the consensus criteria of the Mycoses Study Group and European Organization for Research and Treatment of Cancer. ^b^—Episodes in which voriconazole was used as a suppressive treatment are not included in the effectiveness analysis. ^c^—4 weeks in yeast infection; 6 weeks in mould infection. ^d^—Progression of underlying disease.

**Table 6 jof-07-00456-t006:** Adverse events.

Adverse Events		Severity ^a^	Patients (*N* = 27), *n* (%)
Liver abnormalities	ALT	Grade ≥ 1	13/27 (48.2)
		Grade ≥ 2	7/27 (25.9)
		Grade ≥ 3	1/27 (3.7)
		Grade ≥ 4	-
	TBIL	Grade ≥ 1	8/27 (29.6)
		Grade ≥ 2	6/27 (22.2)
		Grade ≥ 3	2/27 (7.4)
		Grade ≥ 4	-
	ALP	Grade ≥ 1	9/27 (33.3)
		Grade ≥ 2	5/27 (18.5)
		Grade ≥ 3	1/27 (3.7)
		Grade ≥ 4	-
	GGT	Grade ≥ 1	22/27 (81.5)
		Grade ≥ 2	15/27 (55.6)
		Grade ≥ 3	11/27 (40.7)
		Grade ≥ 4	6/27 (22.2)
Visual		Grade 1	1/27 (3.7)
Neurological/Psychiatric		Grade 1	2/27 (7.4)
Gastrointestinal		Grade 2	2/27 (7.4)
Skin		Grade 2	2/27 (7.4)
Renal		Grade 2	1/27 (3.7)

Abbreviations: ALP, alkaline phosphatase; ALT, alanine aminotransferase; GGT, gamma-glutamyl transferase; TBIL, total bilirubin. ^a^—Severity of adverse effects according to the Common Terminology Criteria for Adverse Events v5.0 (US Department of Health and Human Services, National Institutes of Health National Cancer Institute).

**Table 7 jof-07-00456-t007:** Variability of plasma voriconazole levels in recent paediatric studies.

Author (Year of Publication)	Patients / Determinations, Number	Age, Years	Initial Dose	% PC < 1 mg/L	% PC 1–5.5 mg/L	% PC > 5.5 mg/L	Comments
Soler et al. [17] (2012)	30/196	<18	Lower than 2012 recommendations	50%	43%	7%	- Predominance of white Europeans - Median dose for correct PC in <5 years: 38 mg/kg/day
Pieper et al. [29] (2012)	74/251	<18	Lower than 2012 recommendations	57.7%	34.2%	8%	- Predominance of white Europeans - Therapeutic range considered: 1–5 mg/L
Bartelink et al. [30] (2013)	61/380	<20	Lower than 2012 recommendations	1st PC 61%	1st PC 34%	1st PC 5%	- Ethnicity NA (study conducted in The Netherlands) - Therapeutic range considered: 1–5 mg/L - Median dose for correct PC in <2 years: 31.5 mg/kg/day - 80% PC correction when dose adjusted (dose adjustment recommendation not specified)
Choi et al. [49] (2013)	27/193	<19	Lower than 2012 recommendations	31.6%	63.7%	4.7%	- Asian population - Therapeutic range considered: 1–6 mg/L - Mean dose for correct PC in ≤6 years: 17.8 mg/kg/day P.O.
Kang et al. [45] (2015)	31/271	<19	Lower than 2012 recommendations		1st PC 36.6%		- Asian population - 61 patients, 31 of whom underwent VRC monitoring
				Overall 29.9%	Overall 49.4%	Overall 20.7%	
Silva et al. [44] (2016)	26/112	2–18	Lower than 2012 recommendations	46%	47%	7%	- Ethnicity NA (study conducted in Chile) - Relationship between IV route and correct PC
Boast et al. [46] (2016)	55/256	<18	Lower than 2012 recommendations	44.2%	44.2%	11.7%	- Ethnicity NA (study conducted in Australia) - Therapeutic range considered: 1–5 mg/L - Mean dose for correct PC in <6 years: 17.6 mg/kg/day
Kato et al. [47] (2016)	20/111	<18	Lower than 2012 recommendations	1st PC 55%	NA	NA	- Asian population - Median dose for correct PC in ≤5 years: 13.1 mg/kg/day IV and 30.1 mg/kg/day P.O. - Association of sub-therapeutic PCs with oral administration and younger age
Liu et al. [50] (2017)	107/128	<12	Lower than 2012 recommendations	1st PC 48.6%	1st PC 47.7%	1st PC 3.7%	- Asian population - Relationship between omeprazole and PC elevation - Relationship between hypoalbuminemia and PC elevation
Martin et al. [32] (2017)	53/NA	2–17	According to 2012 recommendations	NA	NA	NA	- 45.3% Asian population (58.1% Asian in IA study) - Included patients from 2009 to 2013
Hu et al. [41] (2018)	42/138	2–14	According to 2012 recommendations	1st PC 37.5%	1st PC 50%	1st PC 14.3%	- Asian population - Mean dose for correct PC: 15.4 mg/kg/day IV and 11.2 mg/kg/day P.O. - Mean dose for correct PC in <6 years: 22 mg/kg/day P.O. - 67% PC correction when dose adjusted (dose adjustment recommendation not specified) - Association of PC elevation with IV route and PPI
Allegra et al. [51] (2018)	237/NA	<18	Lower than 2012 recommendations	NA	NA	NA	- Predominance of white Europeans - 63.5% of patients on VRC prophylaxis - Relationship between higher serum creatinine and PC elevation - Relationship between male sex and higher PC - Positive correlation between age and PC in IV route
Lempers et al. [33] (2019)	21/485	<19	Changes throughout the study		1st PC 52.4%		- Predominance of white Europeans - Therapeutic range considered: 1–6 mg/L - Median dose for PC between 1 and4 mg/L in 2–12 years: 24.5 mg/kg/day IV and 25.9 mg/kg/day P.O. - 60% sub-therapeutic PC corrected and 51% supra-therapeutic PC corrected when dose adjusted (dose adjustment recommendation not specified)
				Overall 24.1%	Overall 58.2%	Overall 17.7%	
Duehlmeyer et al. [34] (2021)	45/127	<20	Changes based on provider and patient	NA	45.7%	NA	- Predominance of Caucasians - Included patients from 2010 to 2016 - 53.5% of patients under TDM were on VRC prophylaxis - Initial dose in prophylaxis lower than 2012 recommendations - Therapeutic range considered: 2 to 5.5 mg/L for treatment and >0.5 mg/L for prophylaxis
Valle et al. (2021)	27/229	2–12	According to 2012 recommendations	1st PC 37%	1st PC 55.6%	1st PC 7.4%	- Predominance of white Europeans - Median dose for correct PC in ≤7 years: 21 mg/kg/day - 75% corrected PC when dose adjusted according to protocolled recommendations - Association of less adequate VRC PC with elevated CRP and severe hypoalbuminemia
				Overall 27.5%	Overall 64.2%	Overall8.3%	

Abbreviations: CRP, C-reactive protein; IA, invasive aspergillosis; IV, intravenous; NA, information not available; PC, plasma concentrations; P.O., orally; PPI, proton pump inhibitors; TDM, therapeutic drug monitoring; VRC, voriconazole.

## Data Availability

The data presented in this study are available on request from the corresponding author.

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
