# Peer review of "Voriconazole Use in Children: Therapeutic Drug Monitoring and Control of Inflammation as Key Points for Optimal Treatment"

_jof, 2021, doi:10.3390/jof7060456_

Round 1

Reviewer 1 Report

This is well performed study and well written manuscript providing us with new data which is of value for our clinical practice and insight in how to dose voriconazole in children. There is still a paucity of data in this respect and this manuscript is clearly contributing to an improved insight in the dosing of voriconazole in children.

I only have some minor comments for the authors:

Typo line 50 and 139 and at other places in the text: infra-therapeutic > intra-therapeutic

Table 2, the authors should explain what ‘suppressive treatment’ means.

Line 189: ‘although the difference was only marginally significant’ should be rephrased as being not statistically significant

Line 196; please explain LMM and F

Line 203 and table 3 and 4, there is a discrepancy in p-values for the association with CRP which is said in the text to be statistically signicant, with the table 3 saying NS. The authors might want to explain the different calculations performed for table 3 and 4 as this is not clear from the text and figure legends as it is now.

Table 5 and text if applicable, change ‘stable response’ to ‘stable disease’

Line 308-309 in the discussion; I was wondering if the authors could provide a bit more detail about why the protocol with respect to how to change the dose was not followed and if some figures could be provided if when using the protocol, how often this has led to an adequate PC of voriconazole.

Line 328-331; please specify that this is for the situation when an additional agent is administered to cover a period of inadequate voriconazole PC.

Author Response

Our thanks to the reviewer for the comments and suggestions to improve our manuscript.

Reviewer 2 Report

Dear authors,

thank you very much for the excellent manuscript. It is well written and easy to follow, I only have the minor remarks below.

Abstract

Line 24-25: Please add that an in-house protocol was used.

Introduction

Line 63-69: Please remove your results and conclusions from the introduction of the manuscript.

Results

Line 159-161: Please write Aspergillus in italics.

Table 2:

Line 178-181: For me it is not clear what “4 Aspergillus sp.” means. Do you mean 4 unidentified Aspergillusisolates? Please clarify this point.

Table 3 & 4: For me it is not clear what has been compared here, which means where the “p’s” come from? The quintessence as stated in the text is clear, high CRP and low albumin levels lead to inadequate voriconazole levels. But what was compared to calculate the p in the table remains unclear. Please clarify this in the tables.

Line 216: Please write 37 in superscript.

Discussion

Line 286: Please write Aspergillus in italics.

Author Response

(The authors gave the same response as above.)
